# The Application of Deep Convolutional Neural Networks to Brain Cancer Images: A Survey

**DOI:** 10.3390/jpm10040224

**Published:** 2020-11-12

**Authors:** Amin Zadeh Shirazi, Eric Fornaciari, Mark D. McDonnell, Mahdi Yaghoobi, Yesenia Cevallos, Luis Tello-Oquendo, Deysi Inca, Guillermo A. Gomez

**Affiliations:** 1Centre for Cancer Biology, SA Pathology and the University of South of Australia, Adelaide, SA 5000, Australia; amin.zadeh_shirazi@mymail.unisa.edu.au; 2Computational Learning Systems Laboratory, UniSA STEM, University of South Australia, Mawson Lakes, SA 5095, Australia; Mark.McDonnell@unisa.edu.au; 3Department of Mathematics of Computation, University of California, Los Angeles (UCLA), Los Angeles, CA 90095, USA; efornaci@gmail.com; 4Electrical and Computer Engineering Department, Islamic Azad University, Mashhad Branch, Mashad 917794-8564, Iran; yaghoobi@mshdiau.ac.ir; 5College of Engineering, Universidad Nacional de Chimborazo, Riobamba 060150, Ecuador; jeseniacevallos@hotmail.com (Y.C.); lptelloq@ieee.org (L.T.-O.); deysi_vib@hotmail.es (D.I.)

**Keywords:** deep learning, DCNN, convolutional neural networks, brain cancer, MRI, histology, classification, segmentation

## Abstract

In recent years, improved deep learning techniques have been applied to biomedical image processing for the classification and segmentation of different tumors based on magnetic resonance imaging (MRI) and histopathological imaging (H&E) clinical information. Deep Convolutional Neural Networks (DCNNs) architectures include tens to hundreds of processing layers that can extract multiple levels of features in image-based data, which would be otherwise very difficult and time-consuming to be recognized and extracted by experts for classification of tumors into different tumor types, as well as segmentation of tumor images. This article summarizes the latest studies of deep learning techniques applied to three different kinds of brain cancer medical images (histology, magnetic resonance, and computed tomography) and highlights current challenges in the field for the broader applicability of DCNN in personalized brain cancer care by focusing on two main applications of DCNNs: classification and segmentation of brain cancer tumors images.

## 1. Introduction

Artificial Intelligence (AI)and Machine Learning (ML) methods play a critical role in industrial processes [1,2,3] and biomedicine [4,5,6,7]. By using ML techniques, we can efficiently handle ambiguous and time-consuming biomedical tasks with nearly the same precision as trained specialists. Advancements in deep learning algorithms as a subfield of ML have demonstrated their strength in biomedicine data analysis, particularly in cancer data including patients’ images and clinical information [8,9,10,11].

In the recent decade, many researchers have given special attention to Deep Convolutional Neural Networks (DCNNs) as a very powerful machine-vision tool among deep learning techniques [12]. By applying DCNNs to patients’ X-ray, Computed Tomography (CT), and histopathological images, various types of cancers such as breast [13,14], prostate [15,16], colorectal [17,18], kidney [19,20], and brain [21,22] have been diagnosed in their early stages.

Brain cancer is the first leading cause of mortality among females age 20 and younger, and males age 40 and younger [23]. A large collection of brain cancer patients’ images have been curated and are now rapidly available [24]. Studies show that brain tumors are highly heterogeneous [25] which constitute the main problem for brain tumor classification and segmentation and therefore diagnosis and prognosis. Recently, a very high-quality prospective survey has been done by Muhammad et al. [26] to classify multigrade brain tumors based on Magnetic Resonance (MR) images. In this study, they explored the impact of primary stages such as preprocessing, data augmentation, transfer learning, and different Convolution Neural Network (CNN) architectures on the CNN-based classifiers performance. 

After some inspections, we realized that there is a gap in the application of DCNNs to classify or segment images of brain cancer and most of studies/surveys have just focused on MR or CT images while there is highly-valuable information in the background of histopathological imaging (H&E) images. Therefore, in this literature review, we explore recent advances published between 2018 to 2020 in the use of supervised DCNNs as a robust machine-vision tool in the classification and segmentation tasks of three different kinds of brain cancer images including H&E histology, MR, and CT. For this, articles were searched using the following keywords: “classification”, “segmentation”, “detection”, “brain cancer”, “brain tumor”, “brain tumor lesion”, “brain malignancy”, “brain malignant tissue”, “CNN”, “DCNN”, “convolutional neural network”, “histology”, “pathology”, “histopathology”, “Magnetic Resonance Imaging (MRI)”, “CT”, “imaging”, and “image”. We then assessed the list of identified articles for its content and contribution to the field, in order to include these in this survey. Finally, we also included in this survey some older articles, whose content have contributed key advances in this field. All the works discussed in this survey have used supervised learning to train deep convolutional neural networks.

From the selected studies in the classification and segmentation parts, we found that 32% and 40% of studies have been supervised/validated by a specialist (e.g., pathologists/radiologists), respectively. In these studies, a specialist has been involved in i) dataset preparation or ii) result validation obtained by the methods used in the classification and segmentation parts (for further supervision). In the rest of the articles, the results have been evaluated by testing samples whose results were compared with the ground truths (labels/masks) created by specialists.

Our focus in this work is to provide a legible and concise explanatory paragraph for each recent and relevant work. For this purpose, we did our best to describe the main points in each article in a way that all readers (even the readers who are not very familiar with convolutional neural networks) are able to familiarize with the recent advances in this field. For simplicity, all information provided related to the classification and segmentation tasks have been summarized into two tables. We have also depicted the most important and influential steps of these two tasks in two separated figures as a roadmap for researchers who are going to apply DCNNs in their future work for the classification or segmentation of different kinds of brain cancer images.

Furthermore, for more clarity, the articles in each section were sorted based on the complexity of their methods/algorithms used from simple to complex.

The rest of this article is organized as follows: In Section 2 and Section 3, novel articles applying DCNNs for classification and segmentation tasks to brain cancer images are explored. Indeed, for simplicity, all information for the two tasks mentioned are briefly compiled in two tables. We have also summarized the primary steps and proposed a roadmap for those tasks into separate figures. Finally, in Section 4 and Section 5, we discuss the main contributions of the reviewed works, current limitations and a conclusion summarizing future directions for this rapidly developing field.

## 2. DCNNs Application in the Classification of Brain Cancer Images

In this section, the application of DCNNs to classify brain cancers from different kinds of brain cancer images is explained. For this purpose and to add clarity, we have divided this part into three subsections with each of these focusing on a specific type of images (i.e., H&E histology, MRI, or both).

### 2.1. DCNNs Application in the Classification of Brain Cancer H&E Histology Images

Zadeh Shirazi et al. [24] proposed a new DCNN-based classifier, namely DeepSurvNet, short for “Deep Survival Convolutional Network”, to accurately classify the survival rates of brain cancer patients. The model assigns ranges of survival likelihood across four classifications based on H&E histopathological images. The datasets are collected from the TCGA and a local public hospital, including 450 patients’ H&E slides with different kinds of brain cancer tumors. They considered classes I, II, III, and IV for patients with 0–6, 6–12, 12–24, and more than 24 months of survival after diagnosis, respectively. DeepSurvNet is based on the GoogLeNet [27] architecture and was trained and tested on a public brain cancer dataset from TCGA, and also generalized on a new private dataset. Their model achieved precisions of 0.99 and 0.8 (recalls of 0.98 and 0.81) in the testing stages on the two datasets, respectively. Furthermore, they analyzed the frequency of mutated genes associated with each class, supporting the idea of a different genetic fingerprint associated with patient survival.

Sidong et al. [28] focused on the Isocitrate Dehydrogenase (IDH), an important biomarker in glioma, and predicted its mutational status by using DCNN. Their dataset includes 266 H&E slides gliomas of grade 2 to 4 collected from TCGA and a private hospital. They proposed a model based on using Generative Adversarial Networks (GAN) methodology to generate synthetic but realistic samples to support data augmentation and Resnet50 DCNN architecture as a primary backbone for IDH status prediction. They also concluded that by adding patients’ age as a new feature, the DCNN model can predict IDH status more accurately. They achieved an accuracy of the IDH mutational status of 0.853 (Area Under the Curve (AUC) = 0.927).

Sumi et al. [29] put forward an architecture which can classify four different types of brain tissues of categories healthy, benign, Oligodendroglioma, and Glioblastoma (GBM) based on features extracted from the cellular level (not the tissue level). Their model inputs are H&E histology images of brain tumors. Their model is based on a spatial fusion network where an entire image-wise prediction can discriminate between different sparse features in a whole image. To achieve this, a preprocessing stage extracts patches from each image and applies augmentation. Next, the InceptionResNetv2 (INRV2) architecture is trained on the augmented patches and predicts probabilistic features of different kinds of tumor cancers for local patches. In the second stage, a deep spatial fusion network is implemented, which includes several Fully Connected (FC) and dropout layers to learn spatial relationships between local patches. Finally, a vector calculates the patch tumor class (patch-wise probability). In their work, two datasets, The Cancer Genome Atlas (TCGA) and The Cancer Imaging Archive (TCIA) with 2034 and 2005 basic images have been used, though the training set could be increased with additional data augmentation methods. Additionally, the total patches were extracted, equal to 202,174, and 140,099 from the TCGA and TCIA datasets, respectively. Their model achieved classification accuracy #1 of 0.95 on four-class classification and the classification accuracy #2 of 0.99 on two-class classification (non-necrosis and necrosis).

Yonekura et al. [30] proposed a 14-layer DCNN-based architecture to classify GBM and Low-Grade Glioma (LGG) images. The dataset consists of 200 H&E histological Whole Slide Images (WSIs) from TCGA which contain 100 images. 10,000 distinct patches are extracted from each cohort as inputs to train the DCNN. Additionally, for further performance checking, some popular DCNN architecture such as LeNet [31], ZFNet [32], and VGGNet [33] were trained on those patches and their results were compared with the proposed model. Finally, a classification accuracy of 0.96 for four-fold cross-validation was achieved by the model.

### 2.2. DCNNs Application in the Classification of Brain Cancer MR Images

Fukuma et al. [34] implemented a novel architecture combining several sets of features to identify IDH mutations and TERT promoter (pTERT) mutations. These molecular characteristics are crucial indicators for diagnosing and treating gliomas grades II/III. The model combines patient age, 61 conventional radiomic features, 3 tumor location parameters, and 4000 texture features as input. The texture features are extracted from normalized and cropped tumor lesion slides using AlexNet. Of these feature sets, a subset is selected using F-statistics. To increase the total dataset, augmentation is used. The final accuracy on a nonpublic dataset is 63.1%, which outperforms previous models using only single feature sets.

Chang et al. [35] applied the ResNet50 deep learning model to noninvasively predict IDH1 and IDH2 mutations in glioma grades II-IV. The initial dataset is built from several medical institutions: Hospital of the University of Pennsylvania, Brigham and Women’s Hospital, The Cancer Imaging Center, Dana-Farber/Brigham and Women’s Cancer Center. Skull-stripping, intensity normalization, and tumor extraction preprocessing are applied to the MRI images. The resulting dataset was input into a single combined ResNet50 network that resulted in test accuracies of 87.6%. Additional fields such as age have been shown to improve accuracy by to 89.1%.

Alqudah et al. [36] put forth a new DCNN architecture with just 18 layers for classifying (grading) MR images into three classes of tumors including Meningioma, Glioma, and Pituitary. They used a public dataset containing 3064 brain MR images (T1 weighted contrast-enhanced). In the preprocessing stage, all images are converted into three distinct categories including cropped lesions, uncropped lesions, and segmented lesions. Then, they trained their own DCNN on these three categories and found out that the highest overall performance is related to the uncropped lesions category with 99% accuracy and 98.52% sensitivity.

Kalaiselvi et al. [37] designed and implemented six different DCNN-based classifiers to distinguish LGG and High-Grade Glioma (HGG) tumors from normal lesions through brain cancer MR images. Each model contains just two to five layers. First, each model is trained on the BraTS2013 dataset with 4500 images. The criterion used as a hyperparameter to adjust the models and prevent them from overfitting is early stopping. Then, in the testing phase, they utilized the The Whole-Brain Atlas (WBA) dataset with 281 images. After the training and testing stages, they realized that among the six models, the Five-Layer Model with Stopping Criteria and Batch Normalization (FLSCBN) achieved the lowest 3% False Alarm (FA) and 7% Missed Alaram (MA) (as error rates indexes) and a highest accuracy classification of 89%.

Mzoughi et al. [22] applied a DCNN with 11 layers to 3D whole MR images to classify the grade of glioma tumors into LGG or HGG. In their approach, whole 3D volumetric MRI sequences are passed to the DCNN instead of patch extraction from the MR image. In this study, the dataset used is BraTS2018, which comprises 351 MR T1-weighted images, including mixed grades of glioma tumors. To solve the problem of data image heterogeneity, all images are preprocessed using adaptive contrast enhancement and intensity normalization methods. Furthermore, to generate additional training data, augmentation was used. Based on the results, their proposed 3D DCNN model achieved an overall accuracy of 0.96 on validation data images, which is relatively higher in comparison with other traditional and shallow DCNN approaches.

Badža et al. [38] presented a relatively simple DCNN-based architecture with only 22 layers to classify three brain cancer tumors including Meningioma, Glioma, and Pituitary. The model was trained on a public dataset which contains 3064 T1-weighted contrast-enhanced MR images. With regard to their model inputs being MR whole images than extracted patches, they increased the number of images three times, i.e., to 9192, by using data augmentation techniques (images vertical flipping and 90-degree rotation). The proposed architecture was tested several times and finally found out the best result with the accuracy of 96.56% is related to the conditions that the model is validated by the augmented dataset and 10-fold cross-validation method. Their simple model would be suitable for users who are going to run the network with limited hardware, such as mobile phones or conventional PCs and who want to see the results quickly.

Liu et al. [39] designed and implemented a model to classify three distinct types of brain tumors including Meningioma, Glioma, and Pituitary. The publicly available dataset used in their network is of 3064 T1-weighted contrast-enhanced MR images. Their model is powered by ResNet34 architecture, which uses an all-convolutional architecture with a global pooling layer right before the output. They modified the standard ResNet architecture slightly so that additional global average poling layers were applied to feature maps much closer to the network input. The output of these layers was concatenated to the output global average pooling layer, in order to fuse high and low-level features. They call their network G-ResNet which is short for Global Average Pooling Residual Network. In their experiments, they tested four combinations of concatenation layers between different layers to integrate low-level and high-level features and determine which resulted in the best model accuracy. Indeed, they realized modifying loss function or using the sum of the loss functions such as cross-entropy and interval, the model accuracy would improve. Based on all modifications they applied, their model accuracy could achieve 0.95 as total classification accuracy.

Hemanth et al. [40] proposed a simple DCNN architecture with some modifications in the FC layer. This proposed model considerably reduced computational complexity and is able to classify four brain tumor types of Meningioma, Glioma, Metastasis, and Astrocytoma. T1, T2, and T2 flair MR images are fed as inputs into the model. A total of 220 images were used in their paper, as collected from a private clinic. Their main contribution is to eliminate the updating procedure of weights by using the GD algorithm in the FC layers by replacing them with a simplified approach such that the number of trainable parameters is greatly reduced. They proved that with this amendment, the proposed model can classify brain tumors with an accuracy of 0.96.

Afshar et al. [41] developed a new classifier based on CapsNets to classify three types of brain cancer tumors. To train their proposed network, they used a publicly available dataset including 3064 T1-weighted MR images related to Meningioma, Glioma, and Pituitary tumors. Although CapsNets have some benefits over traditional DCNNs, they are considerably more sensitive to image backgrounds. Hence, preprocessing stages such as tumor segmentation from the whole image and removing backgrounds are highly recommended. However, in this work, the authors reinforced the CapsNets architecture by providing the whole tumor images as the main inputs of their model and the tumor surrounding tissues as extra inputs in its final layer as they considered that tumor segmentation procedure can at the same time introduce disadvantages as it is not only a time-consuming task but also can eliminate some important information such as the tumor boundaries. As a result, their proposed classifier accuracy of 0.91 shows that their developed CapsNet outperforms the previous CapsNets.

Seetha et al. [42] used a transfer learning approach in their DCNN architecture to distinguish brain tumors from nontumors. They applied a DCNN model pretrained on the ImageNet dataset. The public datasets used in their work are Radiopaedia and BraTS2015 containing brain MR images. In their approach, all layers in the pretrained model are frozen, excluding the final layer. The training procedure on the new MR images remains the same in the latest layer. Although this type of classification task only differentiates two classes from each other (i.e., tumor and normal tissues), the model was able to save substantial time in the training phase by using transfer learning method and ultimately achieved a classification accuracy of 0.97.

Pashaei et al. [43] proposed an ensemble model Kernel Extreme Learning Machine-CNN (KE-CNN) combining DCNN and the Kernel Extreme Learning Machine (KELM). In their simple model of just 9 layers, the DCNN extracts features from input images and are fed as input vectors to the KELM as the final layer. In other words, KELM is used as an alternative to FC layers and is responsible for classifying the images. They used this model to classify 3064 T1-weighted MR images from a public dataset. This dataset contains three different kinds of brain tumors including Meningioma, Glioma, and Pituitary. Based on their results, the highest classification accuracy of 0.93 was achieved when they applied the radial base function as a kernel function in the KELM classifier.

Zhou et al. [44] put forward a classifier based on DenseNet and LSTM (DenseNet-LSTM). In this model, features are extracted from MR images with an autoencoder architecture i.e., DenseNet. Those features are then fed to the LSTM structure as inputs for classification. They applied these methods to two distinct public and private datasets for training and evaluating the proposed model. The public dataset includes 3064 brain cancer MR images and three types of tumors containing Gliomas, Pituitary, and Meningiomas. In contrast, the brain tumor types in the private dataset are completely different including Glioma, Meningiomas, Metastatic tumors and normal lesions. This dataset contains 422 MR images. The DenseNet-LSTM was trained and tested on both datasets separately and achieved 0.92 and 0.71 of classification accuracies, respectively.

Mohsen et al. [45] used a DNN with seven layers as their classifier rather than DCNN because of their limited hardware resources. In their proposed model, first, brain tumors/normal lesions are segmented by using the Fuzzy C-means method. Next, the DWT technique extracts the major features from the segmented brain tumor MR images and the features are reduced using the Principal Component Analysis (PCA) method. The extracted features are fed into the next part as the inputs of the classifier. Finally, a DNN architecture is applied to classify three different types of brain tumors including GBM, Sarcoma, and Metastasis along with normal brain lesions. The dataset used in this work is publicly available and comprises 66 T2 weighted brain MR images. Although their proposed model is not using DCNN for feature extraction and classification, the model achieved a classification accuracy and AUC of 0.97 and 0.98, respectively.

Ari et al. [46] proposed different models to classify brain cancer as benign or malignant brain cancer tumors. The dataset used in their work contains 16 patients who have been diagnosed with a brain tumor via T1-weighted MR images screening. First, the images are preprocessed and the background noise is filtered. In the next stage, two different kinds of classifiers such as a six-layer DCNN and ELM local receptive fields (ELM-LRF) were used to classify the tumor types. The results show that the two classifiers have roughly the same classification accuracy of 0.96 and 0.97, respectively where the ELM-LRF achieved slightly better performance. However, the lower accuracy of the DCNN model could be attributed to the limited number of images in the training part.

Suter et al. [47] put forth two different DCNN-based approaches to predict survival rate in GBM patients. Both used the BraTS2018 dataset containing 293 T1, T1c, T2, and T2-Weighted Fluid-Attenuated Inversion Recovery (FLAIR) MR images. To increase the number of images in the training phase, the images were first segmented. Then both the raw and segmented images were fed into the model as inputs. In the first model, a simple DCNN with five blocks was used. In the second DCNN architecture, clinical information such as patients’ age and resection status was added as extra features into the final FC layer of the model. However, the models’ performance results were not satisfactory. When the extracted features from the second DCNN model (as deep features) were combined with additional clinical information such as tumors intensity, location, and shape, and an Support Vector Classifier (SVC) was used as a final predictive model it led to better prediction results. Their final model was called Ensemble SVC and achieved a survival rate prediction with 0.42 accuracy on the test samples.

Banerjee et al. [48] proposed three distinct DCNN architectures trained on MR images to distinguish between HGG and LGG. These models are trained on extracted patches from MRIs, simple slices, and 3D volumetric multiplanar slices and called PatchNet, SliceNet, and VolumeNet, respectively. To maintain a high number of samples in the training phase and assess all proposed models in the testing phase, the scheme leave-one-patient-out was used. The dataset used in this work includes 491 T1, T1c, T2, and FLAIR MR images of HGG and LGG samples collected from the TCIA public data centre. PatchNet is shallower than the other two models and because of the bigger size and complexity of input images in SliceNet and VolumeNet, they are deeper. The results indicate that the deepest DCNN architecture i.e., VolumeNet trained on the 3D volumetric dataset achieves better classification results with a classification accuracy of 0.97 on the test dataset.

Afshar et al. [49] designed and implemented a classifier based on CapsNets to classify three types of brain cancer tumors. The proposed network used a dataset including 3064 T1-weighted MR images related to Meningioma, Glioma, and Pituitary tumors. The CapsNets outperformed traditional DCNNs considering they can better handle overfitting problems caused by insufficient training instances. In this work, the proposed CapsNet was applied to brain segmented regions and brain whole images as their model inputs. Eventually, the CapsNet was determined to perform better on segmented tumors with a classification accuracy of 0.86.

### 2.3. DCNNs Application in the Classification of Brain Cancer H&E Histology and MR Images

Bagari et al. [50] put forward a novel DCCN approach by using both H&E WSIs and MR images to classify low-grade gliomas into categories of Astrocytoma and Oligodendroglioma. The dataset used in this work contains 30 and 20 different patients in the training and testing phase, respectively. All patients have both MR and H&E histological images. First, preprocessing stages such as RoI extraction from H&E histological images, patch extraction, and stain normalization were used. Then, an autoencoder detected abnormal patches among all patches, and finally, a DenseNet-based architecture was applied to the abnormal patches to distinguish Astrocytoma tissues from Oligodendroglioma tissues. Next, a trained 3D DCNN model was applied on MR images including FLAIR, T1, T1C, and T2 MR sequences to segment the tumor lesions. Then, another DCNN architecture extracts the principal features from segmented tumors and feeds them as inputs to a logistic linear regression classifier. Finally, the classifier detects the type of low-grade glioma tumors and combines the results with the previous classifier and results in the total performance of the model. Their combined approach achieved a classification accuracy of 0.9.

All information related to the classification/prediction task mentioned above is summarized in Table 1. We have also depicted in Figure 1 the main steps of this part as a roadmap for researchers who are going to apply DCNNs in their future work for the classification task.

Working and manipulating brain cancer images to recognize the hidden features inside the images and subsequently detect/classify/predict relevant diseases are extremely essential for clinicians, and noninvasive methods including CNN-based classifiers can do this task very well. In general, CNN-based classifiers might be applied to three different kinds of brain cancer images including H&E histology, MR, or both. These classifiers can categorize various brain tumor types or predict biomarker status and patients’ survival rate based on the features inside the images. Hence, two separated and important parts can be considered to achieve the best results: preprocess and CNN architecture. Based on the various studies that we completed, some steps which play important roles in the preprocessing part include patch extraction, abnormal patch detection/selection, intensity normalization, contrast enhancement, and background removing. Overall, there is no best practice to recommend one or several steps in the preprocessing phase and researchers have to try many experiments to find the best results. However, studies show that data augmentation techniques to expand the number of data samples as model inputs have successfully helped the CNNs to achieve higher accuracy. In the second part, CNN models can be first trained to extract features from brain cancer images, and then, those feature vectors along with patients’ clinical data together make a numerical dataset. Afterwards, some linear regression models or nonlinear machine learning methods use this dataset to classify/predict relevant biomarker status, survival rate, or brain tumor types. However, It would be simpler to directly apply CNNs as a classifier or predictive model (Figure 1).

## 3. DCNNs Application in the Segmentation of Brain Cancer Images

In this section, the application of DCNNs to extract the most important features from different kinds of brain cancer images for detection/segmentation task is explained. For this purpose and more simplicity, we have divided this part into three subsections where in each one a specific image type i.e., H&E histology, MRI, or CT, is explored.

### 3.1. DCNNs Application in the Segmentation of Brain Cancer MR Images

Ismael et al. [69] applied the DCNN ResNet50 architecture to segment glioma, meningioma, and pituitary tumor tissues of 3064 MRI images from 233 patients. ResNet50 uses skip connections to avoid gradient degradation, and thereby enables training of much deeper networks than previously thought possible. Due to limited sample in the training dataset, data augmentation was used to generate additional training data to improve results. As the distribution of segmented classes is not uniform, accuracy alone is not a suitable measure of performance. Instead, a combination of accuracy, precision, recall, F1-score, and the balanced accuracy were used, which were 97%, 98%, 97%, 97%, and 97%, respectively, at a patient-level.

Maharjan et al. [70] propose an enhanced softmax loss function. The model was applied to glioma, meningioma, and pituitary tumor segmentation of 3064 MRI images. The resulting loss function utilizes regularization and is far more suitable for multiclass classification than traditional loss functions. This difference helps avoid overfitting when compared to the traditional sigmoid loss function. Using the enhanced softmax loss function, the accuracy was improved to 99.54% 98.14%, and 98.67% for meningioma, glioma, and pituitary tissues, respectively. Regularization also improved runtime by 40–50 ms per sample.

A novel brain tumor segmentation architecture was proposed by Vijh et al. [71] that employs a blend of Otsu thresholding, Adaptive Particle Swarm Optimization (APSO), and morphological operations in the skull stripping preprocessing steps. Skull stripping removes noncerebral tissue not needed for analysis and is a crucial step in neurological imaging. Once preprocessed, 19 features (cluster prominence, cluster shade, contrast, etc.) are extracted from the cerebral tissue image using GLCM (Grey Level Co-occurrence Matrix). These features are passed into a densely connected three-layer CNN. The model yields 98% accuracy applied to the Internet Brain Segmentation Repository (IBSR) dataset, which consists of 18 samples of various tumor types.

Rani et al. [72] propose a method that utilizes adaptive thresholding and high boost convolution filtering to segment brain tumors. Preprocessing steps first extract 2D slices from the 3D MRI scans and any grainy noise is filtered using averaging techniques. The preprocessed dataset is input into the segmentation state which applies Region Growing & Local Binary Pattern (LBP) operators to build a feature vector. This is then fed into the mask generation stage that applies the Fuzzy C-Means Algorithm, Otsu Thresholding, and high boost convolution filtering to extract the high energy glioma regions. The resulting model scored 87.20% for HGG on 210 samples and 83.77 for LGG on 75 samples.

Deng et al. [73] propose a novel architecture to segment FLAIR, Tc1, and T2 images from the BraTS2013 and 2015 datasets of over 270 HGG and LGG scans. The architecture composes HCNN and CRF-RRNN models to segment. The HCNN creates image slices at mixed scales to better leverage location and context at a greater scale. The HCNN model also fuses axial, coronary, and sagittal images. The CRF-RRNN takes the output of the HCNN and produces a global segmentation based on the slices input into the HCNN. The resulting model scored 98.6% accuracy.

Deng et al. [74] improve upon the FCNN network. Batch normalization is added to the network to improve computational performance. The addition of Dense Micro-block Difference features also assist with spatial consistency. Utilizing Fisher vector encoding methods better invariance to texture rotation and scale. The model achieved 91.29% average Dice score on the BraTS2015 dataset on 220 HGG and 50 LGG scans.

Kumar et al. [75] propose a 3D CNN architecture that focuses on correcting intensity inhomogeneity. This is achieved through a preprocessing step that utilizes a novel N3T-spline to correct bias field distortion for reducing noise and intensity variation in the 3D scans. The N3T-spline utilizes the standard N3 (nonuniformity non-parametric normalization) framework but uses a T-spline smoothing strategy. A grey level co-occurrence matrix (GLCM) layer extracts feature vectors from the preprocessed scans. The feature vectors are input into the novel 3D CNN and a simple thresholding scheme is applied to correct false labels and any other undesired noise. The resulting model scored competitively on the BraTS2015 dataset. The dataset consists of 220 HGG scans and 50 LGG scans.

Mittal et al. [76] propose an architecture utilizing Stationary Wavelet Transform (SWT) and Growing Convolution Neural Network (GCNN) to segment neurological images. The model preprocesses input with Wiener filtering and Otsu Thresholding to remove noise and convert to a binary image. A novel skull stripping algorithm is proposed that leverages blob detection and labelling methods to more effectively remove skill, fat, and skin from the regions of interest. Once the images are preprocessed, features are extracted with SWT and classified with a Random Forest implementation. The GCNN then encodes the classified features into a segmented output. The model was able to achieve an Structural similarity (SSIM) score of 98.6% on the BRAINIX dataset of 2457 scans.

Mittal et al. [77] composed a dataset of MRI scans of Glioma, Meningioma, Pituitary, and Negative brain tumor results. The dataset was used to evaluate the performance of several common pre-trained CNN architectures (simple CCNs, VGG, and Xception). Next, several of the pretrained models were fused together in a composite architecture, specifically using simple CNNs, Xception, VGG16, and VGG19. To avoid overfitting, augmented data was utilized in some branches of the architecture, scaling the dataset from 1167 to 5835 samples. The composite architecture achieved an accuracy of 98.89%.

Thillaikkarasi et al. [78] proposed a deep learning model to classify and segment MRI scans. Images are first preprocessed with LoG and Contrast Adaptive Histogram Equalization (CLAHE) filters. Preprocessed scans are fed into a Spatial Gray Level Dependency Matrix (SGLDM) and the following features are generated: contrast, mean, variance, entropy, energy and homogeneity. The feature set is input into a multiclass-SVM (M-SVM) and the MRI scan is classified as abnormal or normal. Abnormal images are then input into a CNN to segment brain tumors from healthy tissues. The resulting model scored 84% accuracy on 40 MRI scans.

Sharma et al. [79] proposed a novel skull stripping algorithm that utilizes Differential Evolution (DE) and Otsu Thresholding. The algorithm first normalizes the input images and applies a Gaussian filter. Next, a global threshold is calculated and iteratively optimized using Otsu Thresholding and DE method. Morphological operations are then applied to extract the neurological tissues. Features are extracted from the preprocessed images with GLCM. The features used in this model are contrast, energy, entropy, correlation, standard deviation. The feature set is used to train the CNN with the Network Fitting Tool (nf-tool). The model is trained on the IBSR dataset, containing 18 samples of various tumors, and the MS-Free dataset, containing 38 tumor free samples. The model is able to achieve 94.73% accuracy.

Kong et al. [80] expand on the traditional UNet architecture and propose the Hybrid Pyramid U-Net model (HPU-Net). Key additions are batch normalization to the downsampling component of the network to help combat vanishing gradient during the training process. This is particularly important for brain tumor segmentation to avoid missing any small lesions that would otherwise be missed. In the upsampling component, bilinear interpolation is used in favour of convolution transposing of convolutional layers as to not add additional parameters. Additionally, semantic and location features are emitted in each upsampling block and combined to capture multiscale information. This effectively creates a feature pyramid to capture brain tumor lesions with multiscale shapes and sizes. The resulting network was trained on BraTS2015 and BraTS2017 on a collective 430 HGG and 145 LGG images. The network achieved 71% and 80% Dice score, respectively.

Benson et al. [81] apply a modified Hourglass Network to brain tumor segmentation. The original Hourglass Network is an encoder–decoder architecture with several residual blocks. Through experimentation, it was determined that using five downsampling layers instead of seven performed better and was computationally less intensive. A single residual block per level was used instead of two. It was also determined that concatenating layers followed by a 1x1 convolution layer outperformed the original element-wise summation, despite additional memory usage. The resulting architecture yielded 92% accuracy on BraTS2018 of 210 HGG and 75 LGG scans.

Zhou et al. [82] propose an ensemble network with several variations on Model Cascade (MC) Net and One-Pass Multi-Task (OM) Net. Modifications made to the MC-Net include an additional series of nested and dense skip layers to improve the feature map coverage of the encoder-decoder architecture. Another MC-Net variation includes adding multiscale contextual information by including inputs at different scales to more effectively extract semantic features at different resolutions. Variations of the OM-Net include making a deeper model by appending an additional residual block to the original OM-Net. Variations to both networks include adding “Squeeze-and-Excitation” (SE) attention mechanisms to adaptively model interdependencies between channel-wise features. This effectively increases sensitivity to informative features and decreases sensitivity to others. Each model and variation were separately trained and then ensembled into a single model. The resulting model achieved 90% accuracy on whole tumor segmentation on the BraTS2018 dataset, consisting of 210 HGG and 75 LGG scans.

Dai et al. [83] implement an ensemble model consisting of a modified U-Net model and a Domain Adaptive U-Net (DAU-Net) model. The modified U-Net uses five residual blocks in both the encoding and decoding components of the network. Group normalization is also applied to add stability given small batches. The DAU-Net is structurally the same as the modified U-Net except instance normalization is applied instead since it is experimentally shown to boost domain adaptation. In total, nine variations of the mentioned models were trained with varying preprocessing techniques and fused together using XGBoost. The model was trained on BraTS2018 consisting of 210 HGG and 75 LGG scans. The resulting ensemble network scored 91% accuracy on whole tumor segmentation.

Kermi et al. [84] proposed a network based on the U-Net architecture. The modified architecture introduces three residual blocks in both the encoding and decoding phases of the architecture. Unlike the original U-Net blocks, each encoding block uses batch normalization and a Parametric Rectified Linear Unit (PReLU). A convolution layer with a stride of two is applied for downsampling. The novel U-Net architecture achieved an 86.8% Dice score on the BraTS2018 dataset consisting of 210 HGG and 75 LGG images.

Mlynarski et al. [85] extend the classic U-Net model to train with “mixed supervision”. Mixed supervision is defined as a dataset with some images fully-annotated (pixel-wise ground truth segmentation) and weakly-annotated (image-level label denoting the presence of a tumor). The model was trained with 210 HGG and 75 LGG fully annotated MRI scans from the BraTS2018 dataset. As a result, the extended U-Net now has an additional subnetwork that performs image-level classification that determines if the scan has a tumor or not. The additional subnet allows the network to exploit additional samples that are only weakly-annotated. The extended model is also expanded to the multiclass problem to segment several classes of interest: nontumor, contrast-enhancing core, edema, nonenhancing core. The proposed model was shown to output a more accurate segmented image when provided weakly annotated samples.

Wang et al. [86] introduce a unique architecture Brain Image-specific Fine-tuning Segmentation (BIFSeg) to address zero-shot learning in medical image segmentation. Zero-shot learning is the machine learning problem when the model encounters classes not observed during training. BIFSeg, based on the P-Net architecture, takes a bounding box as user input localized to the desired tumor core. Once segmented, further optional user input as scribbles can be used to fine-tune the segmentation. When applied to BraTS2015, the model achieved 86.29% Dice score on 220 HGG and 50 LGG scans.

### 3.2. DCNNs Application in the Segmentation of Brain Cancer CT Images

Monteiro et al. [87] applied a 3D CNN architecture to Traumatic Brain Injury (TBI) CT scans to achieve voxel-wise segmentation into multiclass lesions. The methodology was to first train the CNN on CENTER-TB1 Dataset 1 for initial results & weightings. Then, the CENTER-TB1 Dataset 2 was incorporated and some segmentations were manually corrected. The resulting model was then applied to the CQ500 dataset and achieved a 94% AUC accuracy on over 1000 combined CT scans. Though the overall accuracy of this model is slightly lower than other state-of-the-art models, the proposed model has the added ability to distinguish between different lesion types and progression to better understand and personalize care, which is very important in traumatic brain injuries.

### 3.3. DCNNs Application in the Segmentation of Brain Cancer H&E Histology Images

A standard histopathology scan is on the range of 100,000 × 100,000 pixels. This large scale makes training such models very difficult. Xu et al. [88] proposed a unique approach to supporting classification and segmentation on said large-scale brain tumor histopathology. Their architecture extracts 112 × 112 patches from the original image. The set of patches becomes the input to the ImageNet LSVRC 2013 architecture, generating a 4096-dimensional feature vector per extracted patch. Feature vectors are pooled into a linear SVM classifier to produce probability maps to distinguish necrosis from healthy tissue. The resulting architecture yielded 84% accuracy on the MICCAI 2014 dataset, consisting of 35 samples.

All information related to the segmentation task mentioned above is summarized in Table 2. We have also depicted in Figure 2 the main steps of this part as a roadmap for researchers who are going to apply DCNNs in their future work for the segmentation task.

As explored in the several studies, methods for detecting and segmenting brain tumor tissues can take several forms. In general, architectures follow the following three stages: preprocess, segmentation, and postprocess. In the preprocessing stages, several techniques have proven successful including Otsu thresholding, patch extraction, noise reduction, morphological operations, and intensity normalization. Outside of these standard processes, several novel extensions of these standard practices have been developed including variations on Otsu thresholding. Furthermore, during the training segmentation models, data augmentation techniques have been successful to expand limit datasets to make deep learning feasible. Several models and variations on classic models including Xception Net, U-Net, and SVM with very promising outputs result in brain tumor detection and segmentation (Figure 2).

## 4. Discussion

As seen in the many articles that we have reviewed, DCNNs with classification and segmentation architectures have proven to be very powerful and effective when applied to medical imaging in retrospective studies. Specifically, it has been demonstrated that using pretrained models and transfer learning from generic applications to medical imaging applications saves time and achieves better model performance. In particular, when applied to the analysis of histopathological and MR images for brain cancer and is useful as a predictive tool for analysis of patients’ survival and their correlation with cancer genetic signatures. The reviewed literature provides a roadmap for future classification and segmentation methods and provides examples of their different type of applications. Although there is no best practice or specific guidelines to design and implement DCNN-based classification or segmentation models, some studies reviewed have proved that by applying preprocessing stages such as data augmentation, background removing, noise reduction, intensity normalization, and patch extraction, as well as choosing the appropriate architectures as classification and segmentation models, the performance of DCNN models might be dramatically increased. Figure 1 and Figure 2 summarize the most important points recommended by reviewed works to achieve the best results with DCNNs.

## 5. Challenges and Future Considerations

Deep learning techniques in the medical imaging space come with several challenges. Firstly, there is currently no agreed-upon framework or methodology for selecting model hyperparameters, such as weight decay, residual blocks, dropout, number of hidden layers, and image input size. Instead, hyperparameters need to be determined experimentally. Secondly, within current medical imaging, dataset curation requires expensive and time-consuming work to be done by specialists e.g., radiologists and pathologists which may not be possible in every circumstance. The resulting datasets are poor quality and of an insufficient quantity, requiring heavy augmentation to achieve the high number of samples for DCNNs. Moreover, most of the studies are undertaken using retrospective datasets which leave uncertainty about the extent to which these models can be applied to newly generated data and whether this newly generated data can be used for training and testing of these models. Furthermore, it is becoming widely accepted, that at the very early stages of this type of project (i.e., during the model conception), an often strong, interdisciplinary collaboration between medical imaging professionals and machine learning experts is required. Image preprocessing techniques, such as skull stripping and thresholding, are widely used but no standard algorithms or steps have been defined, even for common datasets or types of medical image. This limits the extent to which these procedures can be applied by nondeep learning experts. Finally, DCNN models applied to segmentation and classification analysis of biomedical imaging have not been extensively integrated with all the available clinical metadata. For example, TCGA data sets contain patient demographics data, tumor RNAseq data, survival data, as well as pathology and RNAseq images, but relatively little has been explored in these areas, except for the very few cases detailed in our review.

When considering future directions, despite the advancements of DCNNs, we still face challenges in translating into clinical practice the use of DCNN-based classification and segmentation methods. So far, many studies have shown the powerful capacity of DCNN methods but still, we lack prospective studies that can further validate them. We envisage a future where these methods are incorporated into clinical trials. These could enable the running and testing of different DCNN models (already created) and measure their relative efficacy on tumor classification and segmentation. In this regard, we believe this survey could be an important source of DCNN application to brain cancer images. It could provide a toolbox for the inclusion of such methods in these studies. As we have seen, classification methods have been only been applied to a few groups of brain cancer patients (e.g., IDH and mutant). However, cancer heterogeneity is much more complex, with four subtypes (Proneural, classical, mesenchymal, and neural, although the neural subtype may be nontumor specific) [95]. Therefore, any uncertainty regarding the applicability of DCNN models on nonstratified analysis can be overcome by better designed clinical studies that implement these methodologies. Despite the accuracy of segmentation approaches increasing significantly, this has not been translated to molecular information regarding transcriptomic profiles and genetic signatures associated with different tumor regions. If available, this would permit the determination of the cellular composition of the tumor microenvironment and tumor-stroma interactions that are important in driving cancer progression. Thus, there are still several areas that can be improved to be able to translate deep learning methodologies into clinical medicine. We believe that future studies which address these issues and formulate standard pipelines to deliver performance and accuracy on reliable datasets will be important to define reference standards for testing and validating new DCNN approaches.

## 6. Conclusions

The ability for physicians to quickly and accurately classify and segment brain tumor scans has never been more important. Breakthroughs in deep learning and DCNN architectures applied to biomedical radiology have almost brought these functions into practice.

In this study, we have reviewed the most novel research articles on deep learning applied to medical imaging problems, specifically brain tumor classification and segmentation tasks. The review has resulted in the creation of a roadmap (summarized in Figure 1 and Figure 2 above) for resolving both tasks. This can be leveraged by researchers for implementing models of their own. In addition, Table 1 and Table 2 present a compilation of relevant information, applied techniques, deep learning networks, and DCNN-based models’ performance for the future development of research in this area.

## Figures and Tables

**Figure 1 jpm-10-00224-f001:**
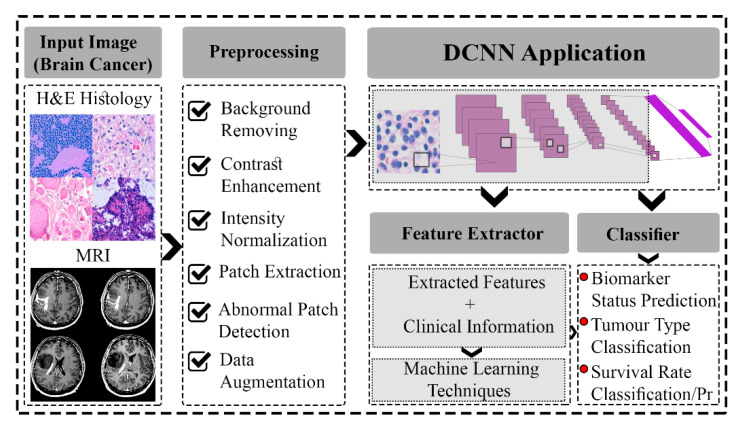
Shows roadmap and stages of a typical brain tumor classification architecture with the following high-level steps: 1. *Input*: Haematoxylin and eosin (H&E) stained histology or magnetic resonance imaging (MRI) can be considered as inputs into the model; 2. *Preprocessing:* apply several techniques to remove background, normalize images, patch extraction and data augmentation; Step 3: *Deep Convolutional Neural Network (DCNN) Application:* The preprocessed dataset is fed into a DCNN model. The model can be used a feature extractor or classify/predict the outputs, directly; if the DCNN model is applied as a feature extractor, these features can be combined with patients clinical information and make a numerical dataset to apply as inputs of machine learning models to classify/predict the outputs (the architecture is now outdated, but is used because its relevant for cited papers); 4. *Model outputs:* brain cancer biomarker status prediction, tumor types classification, or survival rate classification/prediction (Pr).

**Figure 2 jpm-10-00224-f002:**
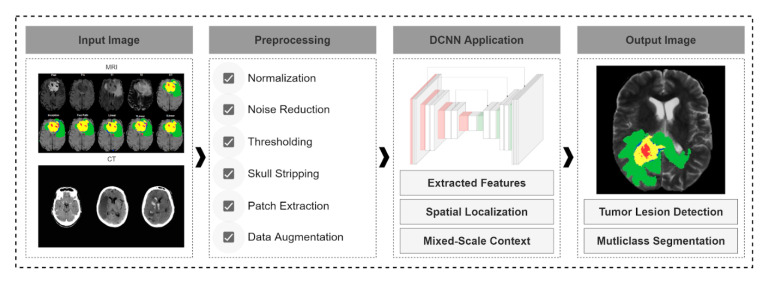
Shows suggested roadmap and stages of a typical brain tumor segmentation Architecture with the following high-level steps: 1. *Input:* magnetic resonance (MRI) and computed tomography (CT) scans are input into the model; 2. *Preprocessing:* apply several techniques to normalize images, remove noise, and filter irrelevant components; Step 3: *Deep Convolutional Neural Network (DCNN) Application:* The preprocessed dataset is fed into a DCNN model the extract features for segmentation, with localization a key component; 4. *Output Images:* Specifies the result of the segmentation model.

**Table 1 jpm-10-00224-t001:** DCNN-based classifiers brief description (sorted by year published/DCNN performance).

Ref.	Year	Task	Tumor Type	Image Type	Model Name	Model Desc.	Software	Hardware	Dataset	Instances/ Cases	Perf.
[26] **	2020	Classification	– Glioma – Meningioma – Pituitary – Glioblastoma	MRI	Six DCNN Architectures *	A prospective survey on Deep Learning techniques applied for Multigrade Brain Tumor Classification	– Caffe – NVidia DIGITS	NVidia TITAN X (Pascal)	– multigrade brain tumor [51] – Brain tumor public data Set [52] – TCIA [53] – BraTS 2015 [54] – Harvard whole-brain Atlas [55] – Internet Brain Segmentation Repository [56]	– 121 – 3064 – 49 – 274 – 30 – 18	– Accuracy: 0.93 (Achieved by VGGNet [33] on multigrade brain tumor [51]) – Accuracy: 0.94 (Achieved by VGGNet [33] on brain tumor public data [52])
[24] **	2020	Classification	– Astrocytoma – Mixed-glioma – Oligodendroglioma – Glioblastoma	H&E Histology	DeepSurvNet	Brain cancer patients’ survival rate classification by using deep convolutional neural network	– Python – TensorFlow – Keras	4xNVidia 1080 Ti GPU	– TCGA [57] – Private dataset	– 400 – 9	– Precision: 0.99 – Precision: 0.80
[28] **	2020	Prediction	– GBM – LGG – Gliomas (Grade II to IV)	H&E Histology	GAN-based ResNet50 *	Gliomas’ IDH status prediction by using the GAN model for data augmentation and Resnet50 as a predictive model	– Python – TensorFlow	N/A	– TCGA [57] – Privatedataset	– 200 – 66	– Accuracy: 0.88 – AUC: 0.93
[36]	2020	Classification	– Glioma – Meningioma – Pituitary	MR (T1 weighted contrast-enhanced)	18 layers DCNN *	Meningioma, glioma, and pituitary tumors classification by using 18 layers DCNN-based model on MR images	N/A	– Intel Core-I7 processor – 16 Gb RAM	Brain tumor public dataset [58]	3064	– Accuracy: 0.99 – Sensitivity: 0.98
[38]	2020	Classification	– Glioma – Meningioma – Pituitary	MR (T1 weighted contrast-enhanced)	22 layers DCNN *	Meningioma, glioma, and pituitary tumors classification by using 22 layers DCNN-based model based on MR images	MATLAB R2018a	NVidia 1050 Ti GPU	Brain tumor public dataset [58]	3064	Accuracy: 0.96
[37] **	2020	Classification	– LGG – HGG	MR (T2 weighted)	FLSCBN	Tumor vs non-tumor classification by using a five layers DCNN-based model on MR images	– Python – TensorFlow – Keras	– Intel Core-I5 processor – 4GB RAM	– BraTS2013 [59] – WBA [60]	– 4500 – 281	– Accuracy: 0.89 – FA: 0.3 – MA: 0.7
[22] **	2020	Classification	– LGG – HGG	MR (T1-Gado or T1-weighted)	3-D DCNN	11 layers 3-D DCNN-based model to classify glioma tumors into LGG and HGG using the T1-weighted MR images	– Python – TensorFlow – Keras	– Intel Core-I7 processor – 19.5 GB RAM – NVIDI1080 Ti GPU	BraTS2018 [61]	351	Accuracy: 0.96
[34] **	2019	Classification	Glioma	MRI	AlexNet; Linear Support Vector Machine	Identify IDH and pTERT mutations using age, radiomic features, and tumor texture features	Caffe	N/A	Not Publicly Available	164	Accuracy: 63.1%
[29]	2019	Classification	– Normal – Benign – Oligodendroglioma – GBM	H&E Histology	INRV2-based deep spatial fusion network *	A mixed DCNN architecture combining InceptionResNetV2 and deep spatial fusion network to classify four different kinds of brain tumors based on H&E images	PyTorch	NVidia 1080 Ti GPU	TCGA [57] TCIA [53]	– 2034 – 2005	– Accuracy#1: 0.95 – Accuracy#2: 0.99
[39]	2019	Classification	– Glioma – Meningioma – Pituitary	MR (T1 weighted contrast-enhanced)	G-ResNet	Meningioma, glioma, and pituitary tumors classification by using a ResNet34-based model with global average pooling and modified loss function based on MR images	PyTorch	NVidia 1080 Ti GPU	Brain tumor public dataset [58]	3064	Accuracy: 0.95
[40]	2019	Classification	– Metastasis – Meningioma – Glioma – Astrocytoma	MR (T1, T2, and T2 flair)	MDCNN	Metastasis, Meningioma, Glioma and Astrocytoma tumors classification using a modified DCNN with reduced computational complexity based on MR images	N/A	N/A	Brain tumor private dataset [62]	220	Accuracy: 0.96
[30]	2018	Classification	– GBM – LGG	H&E Histology	Deep CNN	GBM and LGG classification by using DCNN-based model based on H&E Histological images	– Python – TensorFlow	– NVidia 1080 Ti GPU – Intel Core-I7 processor – 32 GB RAM	TCGA [57]	200	Accuracy: 0.96
[50]	2018	Classification	– Astrocytoma – Oligodendroglioma	H&E Histology; MR FLAIR, T1, T1C, and T2 images	A combined DCNNs-based network *	Astrocytoma and Oligodendroglioma classification by using DCNN-based model based on both MR and Histological images	N/A	N/A	Private dataset	50	Accuracy: 0.90
[43]	2018	Classification	– Glioma – Meningioma – Pituitary	MR (T1 weighted contrast-enhanced)	KE-CNN	Meningioma, glioma, and pituitary tumors classification by using a mixed approach of DCNN and extreme learning based on MR images	N/A	N/A	Brain tumor public dataset [58]	3064	Accuracy: 0.93
[44]	2018	Classification	Public dataset: – Glioma – Meningioma – Pituitary – Private dataset: – Normal – Meningioma – Glioma – Metastasis	– MR (T1 weighted contrast-enhanced); – MR FLAIR, T1, T1C, and T2 images	DenseNet-LSTM	– Meningioma, glioma, and pituitary tumors classification by using DenseNet-LSTM based on MR images – Normal lesion and Meningioma, Glioma, and Metastasis tumors classification by using DenseNet-LSTM based on MR images	– Python – TensorFlow	Nvidia Titan Xp GPU	– Public dataset [58] – Private dataset	– 3064 – 422	– Accuracy: 0.92 – Accuracy: 0.71
[41]	2018	Classification	– Glioma – Meningioma – Pituitary	MR (T1 weighted contrast-enhanced)	CapsNet	Meningioma, Glioma, and Pituitary tumors classification by using a developed CapsNet architecture based on MR images	– Python – Keras	N/A	Brain tumor public dataset [58]	3064	Accuracy: 0.91
[49]	2018	Classification	– Glioma – Meningioma – Pituitary	MR (T1 weighted contrast-enhanced)	CapsNet	Meningioma, Glioma, and Pituitary tumors classification by using a developed CapsNet architecture based on MR images	N/A	N/A	Brain tumor public dataset [58]	3064	Accuracy: 0.86
[42]	2018	Classification	– Tumor (N/A) – Non-Tumor (normal)	MR	Pre-trained DCNN	Brian tumors vs nontumors classification by using a pretrained DCNN based on MR images	Python	N/A	– Radiopaedia [63] – BraTS2015 [64]	N/A	Accuracy: 0.97
[45]	2018	Classification	– Normal – GBM – Sarcoma – Metastasis	MR (T2 weighted)	DWT-DNN *	Normal lesion and GBM, Sarcoma, and Metastasis tumors classification by using DWT-DNN based on MR images	– MATLAB R2015a – WEKA 3.9	N/A	Public dataset [65]	66	– Precision: 0.97 – AUC: 0.98
[46]	2018	Classification	– Benign – Malignant	MR (T1 weighted)	DCNN vs ELM-LRF *	Benign vs malignant tumors classification by using DCNN and ELM-LRF models on MR images	MATLAB R2015a	N/A	Public dataset [66]	16	– Accuracy: 0.96 – Accuracy: 0.97
[47] **	2018	Prediction	GBM	MR (T1-weighted, T1c, T2-weighted, FLAIR)	SVC Ensemble	GBM patient survival rate classification by using two different DCNN models based on MR images	Python	Nvidia Titan Xp GPU	BraTS2018 [61]	293	Accuracy: 0.42
[48]	2018	Classification	– GBM – LGG	MR (T1-weighted, T1c, T2-weighted, FLAIR)	– PatchNet – SliceNet – VolumeNet	GBM and LGG classification by using DCNN-based models based on MR images	– Python – TensorFlow – Keras	– NVidia 1080 GPU – Intel Core-I7 processor – 32 GB RAM	– TCGA-GBM [67] – TCGA-LGG [68] – TCIA [53]	461	Accuracy: 0.97
[35] **	2018	Classification	Glioma	MRI	ResNet50	Identify IDH1/2 mutations in glioma grades II-IV using ResNet50	– Kera – Tensorflow	N/A	From Hospital of the University of Pennsylvania, Brigham and Women’s Hospital, The Cancer Imaging Center, Dana-Farber/Brigham and Women’s Cancer Center	603, 414, 471 (With respect to sources)	Accuracy: 85.7%

* Model names with asterisks are not defined in the original papers and names were assigned based on the models applied. Note: for abbreviations description in this table please refer to the list of abbreviations on the back partof this article (before References). ** the references with “**” mean that the results achieved by their methods or the dataset used have been validated/supervised by specialists (e.g., pathologists/radiologist).

**Table 2 jpm-10-00224-t002:** DCNN-based segmentation models brief description (sorted by year published//DCNN performance).

Ref	Year	Tumor Type	Task	Model Name	Image Type	Model Desc.	Software	Hardware	Dataset	Instances	Performance
[70] **	2020	Glioma, Meningioma, Pituitary	Segmentation	ELM-LRF	MRI	Implemented an enhanced softmax loss function that is more suitable for multiclass applications.	Python 3.6; Keras	– 2.8 GHz Intel Core i7 7th gen processor with 16 GB RAM and 4 GB – NVIDIA 1050 memory	Brain Tumor Dataset [58]	3064	99.54%, 98.14%, 98.67% (Per Tumor Type)
[69]	2020	Glioma, Meningioma, Pituitary	Segmentation	ResNet50	MRI	Glioma, meningioma, and pituitary tumor segmentation with the ResNet50 architecture.	– Python 3.6; Keras 2.2.4; – Tensorflow 1.13	– NVIDIA GeForce RTX 2070 – GPU; Intel i5-9600K @ 3.7 GHz and 16 GB RAM	Brain Tumor Dataset [58]	3064	99%
[73]	2020	Glioma	Segmentation	HCNN; CRF-RRNN	MRI	The composite architecture of HCNN to capture mixed scale context and CRF-RRNN reconstruct a global segmentation.	N/A	N/A	– BraTS2013 [59] – BraTS2015 [64]	220 HGG; 50 LGG	98.6%
[74]	2019	Glioma	Segmentation	FCNN; DMD	MRI	Enhanced FCNN with batch normalization and DMD features to provide spatial consistency. Fisher vector encoding method for texture invariance to scale and rotation.	Caffe	– CPU Intel Core i7 – 3.5GHz, GPU NVIDIA GeForce GTX1070	BraTS2015 [64]	220 HGG; 50 LGG	91%
[86] **	2018	Glioma	Segmentation	P-Net; PC-Net	MRI	Addresses zero-shot learning by taking user input bounding boxes and scribbles to fine-tune segmentations.	Caffe	– 2 8-core E5-2623v3 Intel – Haswell, a K80 NVIDIA GPU and 128GB memory	BraTS2015 [64]	220 HGG; 50 LGG	86.29%
[75]	2019	Glioma	Segmentation	FCNN	MRI	A novel N3T-spline utilizes is used to preprocess 3D input images. GLCM extracts feature vectors and are inputs into a CNN.	MATLAB R2017a	N/A	BraTS2015 [64]	220 HGG; 50 LGG	N/A
[71]	2020	N/A	Segmentation	3-layer DCNN *	MRI	Utilized Otsu thresholding to create a novel skull stripping algorithm. GLCM and a three-layer CNN segments the stripped images.	MATLAB R2018b	N/A	IBSR [89]	18	98%
[72]	2020	Glioma	Segmentation	Automatic Detection and Segmentation of Tumor (ADST) *	3D MRI	Region Growing and Local Binary Pattern (LBP) operators are used to build a feature vector that is then segmented	N/A	N/A	BraTS2018 [61]	210 HGG; 75 LGG	87.20% for HGG; 83.77 for LGG (Average Jaccard)
[81] **	2019	Glioma	Segmentation	Hourglass Net	MRI	Enhanced Hourglass Network with added residual blocks and novel concatenation layers.	N/A	NVIDIA TITAN X GPU	BraTS2018 [61]	210 HGG; 75 LGG	92%
[83] **	2019	Glioma	Segmentation	XGBoost; U-Net; DAU-Net (Domain Adaptive U-Net) *	MRI	Implementation of a U-Net variation using instance normalization to boost domain adaptation.	PyTorch	4 NVIDIA Titan Xp GPU cards	BraTS2018 [61]	210 HGG; 75 LGG	91% (Whole Tumor)
[82]	2019	Glioma	Segmentation	MC-Net; OM-Net	MRI	Ensemble network of several MC-Net and OM-Net variations. Attention mechanisms are added to increases sensitivity to relevant channel-wise interdependencies	N/A	N/A	BraTS2018 [61]	210 HGG; 75 LGG	90% (Whole Tumor)
[84] **	2019	Glioma	Segmentation	U-Net	MRI	The U-Net variation that uses batch normalization and residual blocks to improve performance on neurological images.	N/A	– Intel Xeon E5-2650 CPU@ 2.00 GHz (64 GB) and NVIDIA Quadro – 4000–448 Core CUDA (2 GB) GPU	BraTS2018 [61]	210 HGG; 75 LGG	86.8% (Whole Tumor)
[85]	2019	Glioma	Segmentation	U-Net	MRI	Extension of the U-Net to train with “mixed supervision”, meaning both pixel-wise & image-level ground truths to achieve superior performance.	N/A	N/A	BraTS2018 [61]	210 HGG; 75 LGG	N/A for the entire dataset
[77]	2019	Glioma, Meningioma, Pituitary, and Negative	Classification	– CNN, – Xception, VGG16, VGG19	MRI	Several of the pre-trained models (simple CNNs, Xception, VGG16, and VGG19) were fused together in a composite architecture.	Keras	N/A	N/A	1167	98.89%
[87] **	2020	TBI (Traumatic Brain Injury)	Segmentation	CNN	CT	3D CNN architecture to create voxel-wise segmentation of TBI CT scans.	N/A	N/A	CENTER-TBI (Datasets 1 & 2) [90]; CQ500 [91]	539; 500 (Patients)	94%
[78]	2019	N/A	Segmentation	M-SVM; CNN	MRI	SGLDM and M-SVM are applied to extract and classify MRI scans. CNN is then applied to segment the extracted feature vectors.	N/A	N/A	N/A	40	84%
[76]	2019	N/A	Segmentation	SWT; GCNN	MRI	Dataset is preprocessed with a novel skull stripping. Features are extracted with SWT, classified with a Random Forest implementation and finally segmented with GCNN.	N/A	N/A	BRAINIX [92]	2457	98.6% (SSIM Score)
[80] **	2018	Glioma	Segmentation	U-Net	MRI	HPU-Net enhances the traditional U-Net with multiscale images and image pyramids.	Keras; Tensorflow	NVIDIA Titan X GPU	– BraTS2015 [64] – BraTS2017 [93]	430 HGG; 145 LGG	71% and 80% (Respective to dataset)
[88] **	2015	Glioma	Segmentation	ImageNet LSVRC 2013	H&E Histology	Patches are extracted from large histopathology scans and passed into ImageNet LSVRC 2013 architecture. Linear SVM classifier pools extracted feature vectors.	N/A	N/A	MICCAI 2014 [94]	35	84%

* Model names with asterisks are not defined in the original papers and names were assigned based on the models applied. Note: For abbreviations description in this table please refer to the list of abbreviations on the back part of this article (before References). ** the references with “**” mean that the results achieved by their methods or the dataset used have been validated/supervised by specialists (e.g., pathologists/radiologists).

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
