# Peer review of "The Application of Deep Convolutional Neural Networks to Brain Cancer Images: A Survey"

_jpm, 2020, doi:10.3390/jpm10040224_

Round 1

Reviewer 1 Report

This is a review manuscript focusing on the detection of brain cancer using patient specific clinical images based on deep convolutional neural networks (DCNN). They highlight the recent challenges and discuss future directions. The manuscript is within the scope of the journal but it is not very well written. It needs major revisions. Please see my comments below:

Main issues:
There is no explicit relation, connection between the paragraphs. So, the manuscript looks like presenting a summary paragraph from the selected papers. Is this the aim of the study? If so, it should be explained at the beginning (introduction) of the manuscript.

This is a review paper but there is not any information about the methodology the authors used to prepare the manuscript.
These issues should be clarified.

Is this a narrative, systematic, scoping review or any other type?

How did you decide to choose the papers presented in the manuscript?

Do you have any decision tree to exclude, limit the papers in the literature?

Discussion, Challenges, and future considerations section should be extended. The comments are very general. There should be specific comments about the methodologies, models, preprocessing, etc.

Other issues:

Are there any supervision or not in the studies. It can be clarified.

Figures and Tables: Describe all of the abbreviations used in the figures/tables in the corresponding caption.

There are many abbreviations used in the manuscript but they are not defined at the first usage. For instance:
"By applying DCNNs to patients’ 36 X-ray, CT, and histopathological images, various types of cancers such as breast [13, 14], prostate [15, 37 16], colorectal [17, 18], kidney [19, 20], and brain [21, 22] have been diagnosed in their early stages.", CT
"The datasets are collected from the TCGA and a local public 55 hospital, including 450 patients’ H&E slides with different kinds of brain cancer tumours.", TCGA
"They proposed a model based on using GAN 66 methodology to generate synthetic but realistic samples to support data augmentation and Resnet50 67 DCNN architecture as a primary backbone for IDH status prediction.", IDH

Author Response

Reviewer 1

This is a review manuscript focusing on the detection of brain cancer using patient-specific clinical images based on deep convolutional neural networks (DCNN). They highlight the recent challenges and discuss future directions. The manuscript is within the scope of the journal, but it is not very well written. It needs major revisions. Please see my comments below:

Main issues:

There is no explicit relation, connection between the paragraphs. So, the manuscript looks like presenting a summary paragraph from the selected papers. Is this the aim of the study? If so, it should be explained at the beginning (introduction) of the manuscript.

RESPONSE: Thanks for this valuable comment and we recognize we did not explain it in the text. Yes, our main aim for this survey is to prepare a legible and concise explanatory report (paragraph) of each recent and relevant work. For this purpose, we did our best to describe the essential points in each article in a way that is accessible to a broader audience of readers, including the ones who might not very familiar with convolutional neural networks. Based on your suggestion, we describe the aim of our manuscript in the Introduction section. (Lines 67-74)

“Our focus in this work is to provide a legible and concise explanatory paragraph for each recent and relevant work. For this purpose, we did our best to describe the main points in each article in a way that all readers (even the readers who are not very familiar with convolutional neural networks) are able to familiarize with the recent advances in this field. For simplicity, all information provided related to the classification and segmentation tasks have been summarized into two tables. We have also depicted the most important and influential steps of these two tasks in two separated figures as a roadmap for researchers who are going to apply DCNNs in their future work for the classification or segmentation of different kinds of brain cancer images.

This is a review paper but there is not any information about the methodology the authors used to prepare the manuscript. These issues should be clarified.

Is this a narrative, systematic, scoping review or any other type?

RESPONSE: We appreciate the reviewer in this regard.

This work is a literature (narrative) review/survey. In this novel survey, we have reviewed the most related and recent CNN architectures used in order to (i) classify brain tumours and biological factors based on biomedical/clinical images and (ii) segment brain tumours clinical and histopathological images. Based on your suggestion and to add clarity, we added the following sentence in the manuscript. (Lines 48-53)

“After some inspections, we realized that there is a gap in the application of DCNNs to classify or segment images of brain cancer. Most of studies/surveys have just focused on MR or CT images, however, there is also highly valuable information in the H&E histopathological images, which has been less explored in the literature. Therefore, in this literature review, we explore recent advances published between 2018 to 2020 in the use of DCNNs as a robust machine-vision tool in classification and segmentation tasks of three different kinds of brain cancer images including H&E histology, MR, and CT.”

How did you decide to choose the papers presented in the manuscript? Do you have any decision tree to exclude, limit the papers in the literature?

RESPONSE: Thanks for this valuable comment. We have followed the steps below to select (inclusion criteria), exclude, and explore the articles in the manuscript:

Step 1. The articles with the keywords “classification, segmentation, detection, brain cancer, brain tumour legion, brain malignancy, brain malignant tissue, CNN, DCNN, convolutional neural network, histology, pathology, histopathology, MRI, CT, imaging, image” were explored and downloaded.

Step 2. We primarily focus on the main research articles published between 2018 and 2020, which were extracted from the downloaded articles in step 1. Despite this, we had to include some articles before 2018 since these have contributed some important theoretical concepts to this field. We have also considered only one valuable article published in 2015 that relates to segmentation of brain H&E histopathology images [Xu, Y., et al. Deep convolutional activation features for large scale brain tumour histopathology image classification and segmentation. in 2015 IEEE international conference on acoustics, speech and signal processing (ICASSP). 2015. IEEE.]

Step 3. All the selected articles were then double-checked by the project supervisors, Prof Mark D. McDonnell (computer scientist), Dr Guillermo A. Gomez (Biologist/Brain Cancer Researcher), and Prof Mahdi Yaghoobi (computer scientist), and articles were selected based on the novelty of their methods and relevance.

Step 4. A few articles were removed during writing the manuscript because of their repetitive methods or poor English writing.

Based on your suggestion, we added a short sentence in the manuscript to make this point more clear. (Lines 53-60)

“For this, articles were searched using the following keywords: “classification”, “segmentation”, “detection”, “brain cancer”, “brain tumour”, “brain tumour lesion”, “brain malignancy”, “brain malignant tissue”, “CNN”, “DCNN”, “convolutional neural network”, “histology”, “pathology”, “histopathology”, “MRI”, “CT”, “imaging”, “image”. We then assessed the list of identified articles for its content and contribution to the field, in order to include these in this survey. Finally, we also included in this survey some older articles, whose content have contributed key advances in this field.”

Discussion, Challenges, and future considerations section should be extended. The comments are very general. There should be specific comments about the methodologies, models, preprocessing, etc.

RESPONSE: Thanks for this valuable comment. “Challenges and future directions” are now in a separate section (section 5). Discussion, Challenges and future directions are now extended based on your suggestion: (Lines 623-690)

“4. Discussion,

As seen in the many articles that we have reviewed, DCNNs with classification and segmentation architectures have proven to be very powerful and effective when applied to medical imaging in retrospective studies. Specifically, it has been demonstrated that using pre-trained models and transfer learning from generic applications to medical imaging applications saves time and achieves better model performance. In particular, when applied to the analysis of histopathological and MR images for brain cancer and is useful as a predictive tool for analysis of patients’ survival and their correlation with cancer genetic signatures. The reviewed literature provides a roadmap for future classification and segmentation methods and provides examples of their different type of applications. Although there is no best practice or specific guidelines to design and implement DCNN-based classification or segmentation models, some studies reviewed have proved that by applying pre-processing stages such as data augmentation, background removing, noise reduction, intensity normalization, and patch extraction, as well as choosing the appropriate architectures as classification and segmentation models, the performance of DCNN models might be dramatically increased. Figures 1 and 2 summarise the most important points recommended by reviewed works to achieve the best results with DCNNs.”

“5. Challenges and future considerations

Deep learning techniques in the medical imaging space come with several challenges. Firstly, there is currently no agreed-upon framework or methodology for selecting model hyperparameters, such as weight decay, residual blocks, dropout, number of hidden layers, and image input size. Instead, hyperparameters need to be determined experimentally.  Secondly, within current medical imaging, dataset curation requires expensive and time-consuming work to be done by specialists e.g. radiologists and pathologists which may not be possible in every circumstance.  The resulting datasets are poor quality and of an insufficient quantity, requiring heavy augmentation to achieve the high number of samples for DCNNs. Moreover, most of the studies are undertaken using retrospective datasets which leave uncertainty about the extent to which these models can be applied to newly generated data and whether this newly generated data can be used for training and testing of these models. Furthermore, it is becoming widely accepted, that at very early stages of this type of project (i.e. during the model conception), an often strong, interdisciplinary collaboration between medical imaging professionals and machine learning experts is required. Image pre-processing techniques, such as skull stripping and thresholding, are widely used but no standard algorithms or steps have been defined, even for common datasets or types of medical image. This limits the extent to which these procedures can be applied by non-deep learning experts. Finally, DCNN models applied to segmentation and classification analysis of biomedical imaging have not been extensively integrated with all the available clinical metadata. For example, TCGA data sets contain patient demographics data, tumour RNAseq data, survival data, as well as pathology and RNAseq images, but relatively little has been explored in these areas,  except for the very few cases detailed in our review.

When considering future directions, despite the advancements of DCNNs, we still face challenges in translating into clinical practice the use of DCNN-based classification and segmentation methods. So far, many studies have shown the powerful capacity of DCNN methods but still, we lack prospective studies that can further validate them. We envisage a future where these methods are incorporated into clinical trials. These could enable the running and testing of different DCNN models (already created) and measure their relative efficacy on tumour classification and segmentation. In this regard, we believe this survey could be an important source of DCNN application to brain cancer images. It could provide a toolbox for the inclusion of such methods in these studies. As we have seen, classification methods have been only been applied to a few groups of brain cancer patients (e.g. IDH and mutant). However, cancer heterogeneity is much more complex, with 4 subtypes (Proneural, classical, mesenchymal and neural, although the neural sub-type may be non-tumour specific) [90]. Therefore, any uncertainty regarding the applicability of DCNN models on non-stratified analysis can be overcome by better designed clinical studies that implement these methodologies. Despite, the accuracy of segmentation approaches increasing significantly, this has not been translated to molecular information regarding transcriptomic profiles and genetic signatures associated with different tumour regions. If available, this would permit determination of the cellular composition of the tumour microenvironment and tumour-stroma interactions that are important in driving cancer progression. Thus, there are still several areas that can be improved to be able to translate deep learning methodologies into clinical medicine.  We believe that future studies which address these issues and formulate standard pipelines to deliver performance and accuracy on reliable datasets will be important to define reference standards for testing and validating new DCNN approaches.

Other issues:

Are there any supervision or not in the studies. It can be clarified.

RESPONSE: Thanks for this helpful comment. All the articles studied in the classification and segmentation parts have used supervised learning methods. We added and emphasized this main point in the text as you suggested. (Line 59, 60)

All the works discussed in this survey have used supervised learning to train deep convolutional neural networks.

We think the reviewer refers to “supervision” as a specialist/expert who supervises the result achieved by the method proposed in each article/study. We explored this issue and found that 32% and 40% of studies in the classification and segmentation parts have been supervised/validated by a specialist (e.g. pathologists/radiologists), respectively. In these studies, a specialist has been involved in dataset preparation or result validation obtained by the methods used in the classification and segmentation parts (for further supervision).

In the rest of the articles, the results have been evaluated by testing samples whose results were compared with the ground truths (labels/masks) created by specialists.

Based on your suggestion, we added a description in the manuscript to make this point more clear. (Lines 61-66)

From the selected studies in the classification and segmentation parts, we found that 32% and 40% of studies have been supervised/validated by a specialist (e.g. pathologists/radiologists), respectively. In these studies, a specialist has been involved in i) dataset preparation or ii) result validation obtained by the methods used in the classification and segmentation parts (for further supervision). In the rest of the articles, the results have been evaluated by testing samples whose results were compared with the ground truths (labels/masks) created by specialists.

We have also updated the two tables. In each table, the references with “**” mean that the results achieved by their methods or the dataset used have been validated/supervised by specialists (e.g. pathologists/radiologists).

Figures and Tables: Describe all of the abbreviations used in the figures/tables in the corresponding caption.

RESPONSE: Thanks for this valuable comment. However, the two tables inside the manuscript correspond to the summaries of the whole project and therefore, there are many abbreviations which are not possible to describe in the corresponding captions. Because of this, in the caption, we added a “sentence” to guide readers to search for the abbreviation’s description in the list of abbreviations on the first page.

“Note: For abbreviations description in this table please refer to the list of abbreviations on the first page of this article”

Moreover, for the two figures illustrated in the manuscript, we have now added a description for all abbreviations used in the corresponding Figure captions.

Figure 1: (Lines 399, 401, and 402)

“Figure 1. Shows stages of a typical Brain Tumour Classification Architecture with the following high-level steps: 1. Input: H&E (Haematoxylin and Eosin) stained histology or MRI (Magnetic Resonance) can be considered as inputs into the model; 2. Preprocessing: apply several techniques to remove background, normalize images, patch extraction and data augmentation; Step 3: DCNN (Deep Convolutional Neural Network) Application: The preprocessed dataset is fed into a DCNN model. The model can be used a feature extractor or classify/predict the outputs, directly; If the DCNN model is applied as a feature extractor, these features can be combined with patients clinical information and make a numerical dataset to apply as inputs of machine learning models to classify/predict the outputs (the architecture is now outdated, but is used because its relevant for cited papers); 4. Model outputs: brain cancer biomarker status prediction, tumour types classification, or survival rate classification/prediction (Pr).”

Figure 2: (Lines 612, 614, and 615)

“Figure 2. Shows suggested stages of a typical Brain Tumour Segmentation Architecture with the following high-level steps: 1. Input: MRI (Magnetic Resonance) and CT (Computed Tomography) scans are input into the model; 2. Preprocessing: apply several techniques to normalize images, remove noise, and filter irrelevant components; Step 3: DCNN (Deep Convolutional Neural Network) Application: The preprocessed dataset is fed into a DCNN model the extract features for segmentation, with localization a key component; 4. Output Images: Specifies the result of the segmentation model.”

There are many abbreviations used in the manuscript but they are not defined at the first usage. For instance:
"By applying DCNNs to patients’ 36 X-ray, CT, and histopathological images, various types of cancers such as breast [13, 14], prostate [15, 37 16], colorectal [17, 18], kidney [19, 20], and brain [21, 22] have been diagnosed in their early stages.", CT
"The datasets are collected from the TCGA and a local public 55 hospitals, including 450 patients’ H&E slides with different kinds of brain cancer tumours.", TCGA
"They proposed a model based on using GAN 66 methodology to generate synthetic but realistic samples to support data augmentation and Resnet50 67 DCNN architecture as a primary backbone for IDH status prediction.", IDH

RESPONSE: We appreciate the reviewer in this regard. All the abbreviations used have now been included and sorted alphabetically and available on the first page after abstract. (Line 29)

Reviewer 2 Report

The survey is very short and not comprehensively presented, more work required in all sections.

Section 2 should be divided into subsections, categorization should be made for new readers understanding, only writing long text is not survey. 

The classification and segmentation methods should be categorized. 

The following important literature is missing in the survey: Multi-grade brain tumor classification using deep CNN with extensive data augmentation, Deep learning for multigrade brain tumor classification in smart healthcare systems: A prospective survey

Methods in Table 1 should be year-wise sorted. 

The self discussion should be made on each mainstream method.

Challenges and future directions should be separate sections, where each challenge and future direction should be described in one separate paragraph. 

Author Response

Reviewer 2

The survey is very short and not comprehensively presented, more work required in all sections.

Section 2 should be divided into subsections, categorization should be made for new readers understanding, only writing long text is not survey.

RESPONSE: Thanks for this valuable comment. Sections 2 (classification) and 3 (segmentation), each one is now divided into three subsections based on different kind of images used as follow:

“2.1. DCNNs application in the classification of brain cancer H&E histology images”

“2.2. DCNNs application in the classification of brain cancer MR images”

“2.3. DCNNs application in the classification of brain cancer H&E histology and MR images”

“3.1. DCNNs application in the segmentation of brain cancer MR images”

“3.2. DCNNs application in the segmentation of brain cancer CT images”

“3.3. DCNNs application in the segmentation of brain cancer H&E histology images”

We have also added a brief description in the first part of each section to make this point more clear:

(Lines 84-86)

“In this section, the application of DCNNs to classify brain cancers from different kinds of brain cancer images is explained. For this purpose and to add clarity, we have divided this part into three subsections with each of these focusing on a specific type of images (i.e. H&E histology, MRI, or both).”

(Lines 418-421)

“In this section, the application of DCNNs to extract the most important features from different kinds of brain cancer images for detection/segmentation task is explained. For this purpose and more simplicity, we have divided this part into three subsections where in each one a specific image type i.e. H&E histology, MRI, or CT is explored.”

The classification and segmentation methods should be categorized.

RESPONSE: Thanks for this helpful comment. We have two sections (sections 2 and 3) that differentiate classification vs segmentation methods. Furthermore, for better understanding, in the final part of introduction part, we added a new description and noted that the articles explored in the sections 2 and 3 have been sorted based on the complexity of the methods used. (lines 75,76)

“Furthermore, for more clarity, the articles in each section were sorted based on the complexity of their methods/algorithms used from simple to complex.”

The following important literature is missing in the survey: Multi-grade brain tumor classification using deep CNN with extensive data augmentation, Deep learning for multigrade brain tumor classification in smart healthcare systems: A prospective survey

RESPONSE: We appreciate the reviewer in this regard. This article is now added in the manuscript. (lines 44-47)

“Recently, a very high-quality prospective survey has been done by Muhammad, et al [26] to classify multigrade brain tumours based on MR images. In this study, they have explored the impact of primary stages such as pre-processing, data augmentation, transfer learning, and different CNN architectures on the performance of the CNN-based classifier.”

Methods in Table 1 should be year-wise sorted.

RESPONSE: Thanks for this valuable comment. Table 1 is now year-wise sorted (like table 2 for segmentation methods).

The self-discussion should be made on each mainstream method.

RESPONSE: Thanks for this valuable comment. The self-discussion part is now added on each mainstream method as follow:

1) In the final part of section 2, classification methods: (lines 378-396)

“Working and manipulating brain cancer images to recognize the hidden features inside the images and subsequently detect/classify/predict relevant diseases are extremely essential for clinicians and non-invasive methods including CNN-based classifiers can do this task very well. In general, CNN-based classifiers might be applied to three different kinds of brain cancer images including H&E histology, MR or both. Theses classifiers can categorize various brain tumour types or predict biomarker status and patients’ survival rate based on the features inside the images. Hence, two separated and important parts can be considered to achieve the best results: Pre-process and CNN architecture. Based on various studies we did, some steps which play important roles in the pre-processing part include patch extraction, abnormal patch detection/selection, intensity normalization, contrast enhancement and background removing. Overall, there is no best practice to recommend one or several steps in the pre-processing phase and researchers have to try many experiments to find the best results. However, studies show that data augmentation techniques to expand the number of data samples as model inputs have successfully helped the CNNs to achieve higher accuracy. In the second part, CNN models can be first trained to extract features from brain cancer images and then, those feature vectors along with patients’ clinical data together make a numerical dataset. Afterwards, some linear regression models or non-linear machine learning methods use this dataset to classify/predict relevant biomarker status, survival rate, or brain tumour types. However, It would be simpler to directly apply CNNs as a classifier or predictive model (Figure 1).”

2) In the final part of section 3, segmentation methods: (lines 596-605)

“As explored in the several studies, methods for detecting and segmenting brain tumour tissues can take several forms. In general, architectures follow the following three stages: pre-process, segmentation, and post-process. In the pre-processing stages, several techniques have proven successful including Otsu thresholding, patch extraction, noise reduction, morphological operations, and intensity normalization. Outside of these standard processes, several novel extensions of these standard practices have been developed including variations on Otsu thresholding. Furthermore, during the training segmentation models, data augmentation techniques have been successful to expand limit datasets to make deep learning feasible. Several models and variations on classic models including Xception Net, U-Net, and SVM with very promising outputs result in brain tumour detection and segmentation (Figure 2).”

Challenges and future directions should be separate sections, where each challenge and future direction should be described in one separate paragraph. 

RESPONSE: Thanks for this helpful comment. “Challenges and future directions” are now in a separate section (section 5), and each one has been individually explained in one paragraph

Reviewer 3 Report

In this article, the authors summarize the latest studies of deep convolutional neural network (DCNN) techniques applied to brain tumor (BT) images for classification and segmentation of BTs. In addition, they discuss current limitations and future directions in the applicability of DCNN for precision medicine of BTs.

I have some comments.

As described in the text, determination of molecular characteristics of BTs prior to surgical intervention is thought to be beneficial for tumor diagnosis and patient care, especially in clinical practice of diffuse gliomas. A recent study by Chang et al. applied DNN to MR images of WHO grade II to IV gliomas and demonstrated that IDH1/2 mutations can be successfully classified from MR images.

Chang, K. et al. Residual convolutional neural network for the determination of IDH status in low- and high-grade gliomas from MR Imaging. Clin. Cancer Res. 24, 1073–1081, https://doi.org/10.1158/1078-0432.CCR-17-2236 (2018).

Moreover, identification of IDH mutation and TERT promoter (pTERT) mutation is crucial for reaching a correct diagnosis and choosing the most appropriate treatment for patients harboring WHO grade II/III gliomas. Fukuma and his colleagues sought to predict the molecular characteristics of WHO grade II/III gliomas.

Fukuma, R. et al. Prediction of IDH and TERT promoter mutations in low-grade glioma from magnetic resonance images using a convolutional neural network. Sci Rep 9, 20311 (2019). https://doi.org/10.1038/s41598-019-56767-3

It would be better to summarize these challenges in the text.

Author Response

Reviewer 3

In this article, the authors summarize the latest studies of deep convolutional neural network (DCNN) techniques applied to brain tumor (BT) images for classification and segmentation of BTs. In addition, they discuss current limitations and future directions in the applicability of DCNN for precision medicine of BTs.

I have some comments.

As described in the text, determination of molecular characteristics of BTs prior to surgical intervention is thought to be beneficial for tumor diagnosis and patient care, especially in clinical practice of diffuse gliomas. A recent study by Chang et al. applied DNN to MR images of WHO grade II to IV gliomas and demonstrated that IDH1/2 mutations can be successfully classified from MR images.

Chang, K. et al. Residual convolutional neural network for the determination of IDH status in low- and high-grade gliomas from MR Imaging. Clin. Cancer Res. 24, 1073–1081, https://doi.org/10.1158/1078-0432.CCR-17-2236 (2018).

Moreover, identification of IDH mutation and TERT promoter (pTERT) mutation is crucial for reaching a correct diagnosis and choosing the most appropriate treatment for patients harboring WHO grade II/III gliomas. Fukuma and his colleagues sought to predict the molecular characteristics of WHO grade II/III gliomas.

Fukuma, R. et al. Prediction of IDH and TERT promoter mutations in low-grade glioma from magnetic resonance images using a convolutional neural network. Sci Rep 9, 20311 (2019). https://doi.org/10.1038/s41598-019-56767-3

It would be better to summarize these challenges in the text.

RESPONSE: We appreciate the reviewer in this regard. These two works mentioned above are now summarized and inserted in the manuscript.

  • Fukuma, et al [34]: (lines 130-137)

“Fukuma et al [34] implemented a novel architecture combining several sets of features to identify IDH mutations and TERT promoter (pTERT) mutations. These molecular characteristics are crucial indicators for diagnosing and treating gliomas grades II/III. The model combines patient age, 61 conventional radiomic features, 3 tumour location parameters, and 4000 texture features as input. The texture features are extracted from normalized and cropped tumour lesion slides using AlexNet. Of these feature sets, a subset is selected using F-statistics. To increase the total dataset, augmentation is used. The final accuracy on a non-public dataset is 63.1%, which outperforms previous models using only single feature sets.”

  • Chang, et al [35] (lines 138-144)

“Chang et al [35] applied the ResNet50 deep learning model to noninvasively predict IDH1 and IDH2 mutations in glioma grades II-IV. The initial dataset is built from several medical institutions: Hospital of the University of Pennsylvania, Brigham and Women’s Hospital, The Cancer Imaging Center, Dana-Farber/Brigham and Women’s Cancer Center. Skull-stripping, intensity normalization, and tumour extraction preprocessing are applied to the MRI images. The resulting dataset was input into a single combined ResNet50 network that resulted in test accuracies of 87.6%. Additional fields such as age have been shown to improve accuracy by to 89.1%.”

Table #1 is also updated based on the new studies added.

Round 2

Reviewer 1 Report

The manuscript has been substantially improved by the revision. Thanks to the authors for addressing all of my comments.

Author Response

We appreciate the reviewer is satisfied with the revised version of our manuscript. We thanks again for his/her valuable feedback on the original version.

Reviewer 2 Report

My other comments are addressed well, however, I recommended a recent research article known as "Multi-grade brain tumor classification using deep CNN with extensive data augmentation" is not been included in the surveyed methods Table. This method should be included in the classification-based methods Table. 

Author Response

Thanks for this helpful comment and we apologize for ommiting this article in the table. This article is now included in the first row of table 1.

Reviewer 3 Report

The authors respond to the reviewer's comments.

Author Response

(The authors gave the same response as above.)
